# Understanding teamwork experiences of neurodivergent students: A phenomenological exploration of conflict and collaboration in engineering teams

McKenna P. Sperry*, Thomas A. O'Neill

Department of Psychology, University of Calgary, Calgary, Alberta, Canada

* mckenna.sperry@ucalgary.ca

## Abstract

Neurodivergent individuals have been systematically excluded from employment opportunities, meaning that we know little about lived experiences within neurodiverse teams. To fill this gap, the current phenomenological study seeks to understand the teamwork experiences of neurodivergent students at the University of Calgary. We conducted semi-structured interviews with students in the faculty of engineering who self-identify as neurodivergent, exploring their perceptions and experiences of teamwork and conflict. We grouped the themes that emerged from the interview data into cognitive and behavioural dimensions to represent what neurodivergent students *think* and *do* as they face and manage challenges and opportunities within team settings. Together, the themes highlight that students' desire to be understood and recognized by others significantly influences their team interactions. These findings suggest that traditional theories attempting to explain the social elements of neurodivergence are limited in their capacity to capture the nuanced ways in which neurodivergent individuals seek to understand and relate to others, particularly in team contexts. Finally, this study translates students' insights into recommendations for educators to create more supportive and inclusive teamwork environments in courses.

## Introduction

Team-based learning is pervasive across educational programs. Many accreditation bodies include teamwork as a key learning outcome, emphasizing the importance of preparing students for professional collaboration in their careers [1]. Consequently, understanding how individuals navigate team environments is essential to supporting them in developing teamwork skills. Countless studies have been conducted on factors that underlie effective teamwork in higher education; however, the teamwork experiences of neurodivergent students have remained largely unexplored. Despite

**Data availability statement:** The interview data collected for this study contains potentially identifying and sensitive personal information. Our ethics approval does not permit public data sharing, nor did participants consent to it. Additional questions can be directed to the University of Calgary Conjoint Faculties Research Ethics Board (cfreb@ucalgary.ca). In accordance with PLOS One's data availability policy for studies where data sharing could compromise participant privacy, we provide evidentiary quotes throughout the Results section to support transparency.

**Funding:** This study received funding from the Government of Canada Social Sciences and Humanities Research Council and Schulich School of Engineering at the University of Calgary. The funders had no role in study design, data collection and analysis, decision to publish, or preparation of the manuscript.

**Competing interests:** The authors have declared that no competing interests exist.

individuals with conditions that fall under the neurodivergent umbrella making up 22% of the global population [2], they have remained relatively underrepresented in organizational research until recently [3,4].

Since post-secondary education represents a critical pipeline to the workforce, understanding the experiences of neurodivergent students is an important first step in preparing them for future careers. Emerging research on the institutional experiences of neurodivergent individuals, more broadly, suggests that they experience more stigma and barriers to participation [5,6]. Understanding these experiences could benefit educators in designing courses, facilitating group work, and supporting students in developing teamwork skills that future employers will expect of professionals entering the workforce. So, the question remains: How do neurodivergent students experience teamwork, and how should educational practices evolve in response?

### Defining "neurodiversity"

Singer, as cited in Dwyer (7), posited that disability cannot be traced to a single factor and, instead, is a result of a complex interplay of biological and social factors which can create challenges and opportunities. Researchers tend to agree on the complexity of "neurodiversity"; it simultaneously describes differences in biology and the social climate these differences are embedded in, and its meaning is constantly evolving [7,8]. Despite this definitional controversy, there is consensus regarding the importance of using respectful, inclusive language. According to social information processing theory [9], individuals form attitudes and beliefs based on the information in their environment, mainly through interpreting value-laden social cues. This process influences how individuals construct perceptions of others, including marginalized groups, highlighting the importance of language awareness. Respectful language conveys value and acceptance towards individuals, unlike terms (e.g., "deficient") that imply a negative value judgement [10].

Researchers have used a variety of terms to describe the spectrum of neurocognitive abilities and differences, such as "neurominority," "neurodivergent," and "neuroatypical" [11–13]. For the sake of simplicity and consistency in this paper, we use "neurodiverse" to describe behavioural manifestations of variations in neurocognitive abilities and "neurodivergent" as an umbrella term to refer to an individual or group whose patterns of interaction deviate from social norms due to neurocognitive differences. Specifically, this study focuses on a subset of traits and diagnoses, including autism spectrum disorder (ASD) and attention deficit hyperactivity disorder (ADHD).

### Positioning neurodiversity within the post-secondary context

Accommodation and the duty to accommodate [14] have resulted in more neurodivergent students entering and completing post-secondary programs [15,16]. However, previous findings suggest that neurodivergent students experience less enjoyment and success than their neurotypical peers [17,18]. Post-secondary institutions prioritize standardized communication and evaluation methods based on neurotypical norms, which might require neurodivergent students to suppress unique practices, interactive styles, and strengths that are core to their self-identity [19]. By embedding

neuro-normative ways of thinking and doing, post-secondary environments perpetuate structural stigma through the notion that neurodivergent individuals are deviant [20]. Structural stigma focuses on the macro-level factors that exacerbate a stigmatized status, such as institution-level norms, attitudes, and policy [21,22].

Importantly, structural stigma can trickle into micro-level mechanisms and impact interpersonal interactions and how individuals perceive themselves [22]. Sasson et al. [23] found that non-autistic university students were quick to form negative first impressions of autistic students. However, more encouragingly, Hu et al. [24] found that structured educational training was linked to higher levels of autism knowledge and lower stigma Furthermore, Mellifont [25] described faculty hesitancy in discussing their neurodivergent experiences, as neurocognitive differences are perceived as incompatible with academic pursuits. By reinforcing negative perceptions of neurodivergence, structural stigma might explain why many students go to great lengths to conceal their traits [26]; there are many consequences that accompany being perceived as deviant [20].

Given that status loss and discrimination are behavioural manifestations of stigmatization [27], the goal of engaging in identity management behaviours is to minimize the damaging consequences of stigmatization to one's self-concept and social status [28]. Nearly three-quarters of autistic individuals report "masking" and "camouflaging" to make their autistic traits less noticeable [29]. According to Miller et al. [28], masking is a form of identity management driven by stigma avoidance. Masking and camouflaging are linked to exhaustion, burnout, anxiety, depression, and reduced well-being [19,29–32]. In interviewing autistic individuals about their motives for masking, Cage and Troxell-Whitman [29] found that the most common reason was "fitting in and passing as neurotypical."

## Understanding team conflict

Conflict is an inherent aspect of teamwork and has the potential to push teams towards innovation or drive them apart due to interpersonal tensions; in this sense, conflict is both an opportunity and a challenge [33–35]. As teams begin to work together on an issue or project, conflict arises as incompatibilities between members on how the issue should be approached emerge [34]. How conflict is addressed and managed has significant implications for team effectiveness and member well-being [35,36].

Jehn [37] introduced a conflict model that is widely referenced in conflict literature and includes three distinct conflict types: task conflict, relationship conflict, and process conflict. Task conflict is characterized by perceived incompatibilities between ideas and opinions related to tackling the task, whereas relationship conflict involves heightened emotionality and tension due to perceived interpersonal compatibilities; finally, process conflict results from perceived incompatibilities in how the task should be executed [33,37]. Since task conflict promotes information pooling, taking time to understand team members' perspectives, and analyzing multiple courses of action, it is associated with higher levels of team effectiveness [35].

While this framework provides an essential starting point for understanding how conflict type can influence team-level outcomes, it does not detail processes by which team members influence the team's capacity for constructive or destructive conflict. O'Neill et al. [34] conducted a study that measured teams' levels of all three conflict types and found evidence for the existence of four stable, distinct conflict state profiles. Conflict states, defined as the "shared perception among members of the team about the intensity of disagreement," and the conflict management processes team members participate in interact reciprocally [38]. States influence what conflict management style team members employ by setting the norms and expectations for team conflict, and, in turn, conflict management styles stabilize into states [34,38]. Given this relationship, it is reasonable to extrapolate that changing the conflict management tendencies of team members would be a viable route for influencing conflict states.

Although the previous paragraph highlights that individuals' conflict management styles (CMS) and subsequent behavioural repertoires are not immune from team-level conflict states, they can be a valuable diagnostic source. Rahim and Bonoma [39] proposed that assessing CMS is necessary in order to develop interventions to manage conflict. Most

CMS assessments are grounded in the dual concern model, which posits that conflict management is a function of high or low concern for self and high or low concern for others [40,41]; five distinct styles describe how much an individual varies on two axes that represent concern for self and others (Table 1). Although CMS assessments give a snapshot of someone's conflict management tendencies at a certain point in time, they can be a useful starting point for understanding how the conflict process unfolds. Understanding tendencies to express conflict may provide insight into the complex processes of conflict escalation or de-escalation [40].

## Navigating conflict in neurodiverse student teams

Post-secondary institutions are increasingly finding opportunities to integrate collaborative group projects into coursework since they prepare students for teamwork experiences in their careers. Exposure to various teamwork opportunities is important because the associated skills and competencies are highly transferable to professional environments and sought after by employers [1]. Modern teams are increasingly expected to work across disciplines, distance, and culture [42].

Student and workplace teams are diverse in demographic characteristics, as well as perspectives, expertise, and neurocognitive abilities. The risk of negative conflict is greater in diverse teams, especially where there are deep-level differences; however, the innovation potential is also greater [43,44]. This "high-risk, high-reward" dilemma suggests that how conflict evolves in diverse teams matters. Zolyomi et al. [45] explored the technology needs of neurodiverse teams and found that autistic students may experience challenges such as sensory overstimulation, emotional overwhelm, and a desire for structure that can cause tensions. Furthermore, half of the participants expressed hesitancy about bringing up challenges and preferred to adapt to the team norms and expectations instead of asking team members to accommodate their needs. Other research corroborates findings related to communication and social challenges within the neurodivergent community [46,47].

**Table 1. Construct Definitions Developed for ITP Metric's CMS Instument.**

| Style | Definition |
|---|---|
| Avoiding | Avoiding involves behaviors focused on nonconfrontation, such as delaying, deflecting, or sidestepping the issue, and/or withdrawing from, postponing, or stalling the conflict. |
| Accommodating | Accommodating involves behaviors focused on yielding, or giving into, the interests, preferences, or demands of the other party. Accommodating is cooperative in a nonconfrontational way that prioritizes an acceptance of the other's desires. There is little to no significant attempt to communicate or advance one's own preferences, especially if they are potentially incompatible with the other's preferences. |
| Compromising | Compromising involves behaviors focused on reaching a mutually acceptable settlement that involves both parties making concessions to achieve a "good enough" solution. Importantly, the settlement represents a suboptimal solution that only somewhat satisfies both parties' preferences, lying part way between their preferred outcomes. |
| Integrating | Integrating involves behaviors focused on reaching an optimal agreement by considering both parties' interests and incorporating both parties' preferences. Integrative behaviors involve a cooperative approach, such as seeking to understand each other's interests, joint problem-solving of issues, and explicitly incorporating both parties' preferences to arrive at a solution that fully satisfies each other's needs. |
| Dominating | Dominating involves behaviors focused on persuading, pressuring, or forcing others to accept one's own preferences and goals, at the expense of, or at least inattention to, the other's interests. Dominating behaviors may involve "soft" tactics that seek to influence or persuade others to adopt one's own perspective, or "harder" tactics involving one-sided attempts to overpower others, such as by imposing one's own views onto others. |

Reprinted from O'Neill TA, Aldana AJ, Espinoza JA, Sperry MP, Brykman KM, and Donia MBL. Conflict management styles assessment and feedback for student self-awareness and team development. Assess Eval High Educ. 2024 Nov 16;49(8): 1092. Copyright 2024, reprinted by permission of Informa UK Limited, trading as Taylor & Francis Group, https://www.tandfonline.com.

These findings suggest that neurodiverse teams can experience challenges relating to each other due to their different intersubjective experiences. Milton [48] coined this phenomenon the double empathy problem and described how "different dispositional outlooks and personal conceptual understandings" can increase the potential for bi-directional misunderstandings between neurodivergent and neurotypical individuals. Similarly, representational gaps (rGaps) introduced by Cronin and Weingart [49] suggest that differences in knowledge frameworks and perspectives, more broadly, can result in misunderstandings due to misaligned and incompatible assumptions. The double empathy problem and rGaps align with Zolyomi et al.'s [45] findings, which emphasize that social information is not readily accessible to autistic students, and neurotypical team members are unsure how to bridge the communication gap. Furthermore, when neurotypical team members do not take time to understand the perspectives of their neurodivergent team members, they may inadvertently contribute to stigmatization through negative bias [23]. The double empathy problem inspires investigation into neurodiverse team conflict from an under-recognized perspective, focusing on how neurodivergent individuals interact with the unspoken social norms of educational institutions.

Ultimately, we know little about conflict in neurodiverse teams. Understanding conflict management tendencies and processes of neurodivergent students within neurodiverse teams could provide insight into how their teamwork experiences differ from those of their neurotypical peers. Educators play an integral role in designing, facilitating, and supporting course-based teamwork experiences that students will carry into their careers. As such, educators should consider how institutional norms perpetuate barriers to participation [50] and explore ways to promote more accessible, inclusive, and equitable learning experiences [16,51]. For example, course designers might reflect on how teamwork competencies are *typically* taught and assessed and how alternative approaches might better recognize diverse strengths and reduce stigma. Efforts to make course-based teamwork learning more accessible and inclusive for neurodivergent students may improve the experiences of all students, regardless of neurotype [52].

## Current study

Considering the limited research on neurodiverse teams, there is a need to explore and better understand neurodivergent students' teamwork experiences. Information gathering and exploration are necessary first steps in understanding how teamwork experiences differ across neurotypes and developing theories and practical frameworks to guide educators in supporting neurodiverse teamwork. The current phenomenological study aims to understand how neurodivergent individuals make sense of and interact in teamwork settings by exploring and analyzing the lived team conflict experiences of engineering students at the University of Calgary who identify as neurodivergent.

## Method

### Research setting

This study was conducted within the Faculty of Engineering at the University of Calgary because the group project components of first-year courses are designed to emulate teamwork and collaboration in industry. Beyond testing technical skills, these projects encourage team members to communicate, problem-solve, and adapt to changing needs and circumstances. For these reasons, we anticipated that this theoretically sampled group of participants would have lived experience and an understanding of what it is like to experience conflict and other interactions within student teams.

### Participants and procedure

We recruited 16 participants enrolled in a first-year professional development engineering course and across various undergraduate engineering courses. All students completed a conflict management styles (CMS) assessment through the ITP Metrics online platform (www.ITPmetrics.com). We embedded a question into the survey that asked students if they self-identify as neurodivergent, which, for the sake of this study, we defined as "traits and diagnoses that correspond with autism spectrum disorder (ASD) or attention deficit hyperactivity disorder (ADHD)." Students met the eligibility criteria

solely based on this self-identification item; no clinical verification or diagnostic documentation was obtained. Including participants who self-identify as neurodivergent is a consistent practice within neurodiversity research [53,54]. A large portion of individuals with neurodivergent traits and features are undiagnosed [55,56] likely due to the barriers associated with seeking and obtaining a diagnosis. Furthermore, given that phenomenological research seeks to understand how individuals interpret and make meaning of their lived experiences, we prioritized participants' self-understanding over a clinical threshold. Excluding students without a formal diagnosis would unnecessarily narrow the sample and risk not capturing the breadth of neurodivergent experiences as they relate to teamwork. If the student answered "yes" to this self-identification question, a second question asked whether they would like to be contacted to participate in a study on how neurodivergent individuals make sense of and interact in team conflict.

McKenna Sperry conducted all interviews for sake of consistency. She emailed students who indicated their desire to be involved in the study and invited them to participate in a 45-minute semi-structured virtual interview through Zoom. To increase the accessibility of the interview, she sent interview topics beforehand and asked students if there were additional steps she could take to make the interview more comfortable for them at the beginning. Data collection was covered by Ethics ID REB23–1686, issued by the University of Calgary Conjoint Faculties Research Ethics Board (CFREB). At the start of the interview, McKenna reviewed the study information letter, which was sent to students when they booked their interview, and detailed what they would be asked to do, risks, benefits, confidentiality, and other relevant information. She then documented their verbal consent. Before asking interview questions, she let students know they could ask clarifying questions, take a break, provide feedback, skip questions, or conclude the interview at any point. Interviews occurred from March 7, 2024, to January 31, 2025, and were recorded. Once McKenna had transcribed each interview, she deleted the corresponding recording. She anonymized interview data during transcription.

16 engineering students, including those who self-identify as neurodivergent or have a formal diagnosis, completed the CMS assessment and participated in an interview. These students were first-year through fourth-year undergraduates ($M_{age} = 20.56$, $SD_{age} = 1.55$). 25.0% ($n = 4$) of students identified as female, 62.5% ($n = 10$) as male, and 12.5% ($n = 2$) identified as nonbinary. In terms of neurodivergent identity, 18.8% ($n = 3$) identified with ASD, 56.3% ($n = 9$) identified with ADHD, 12.5% ($n = 2$) identified with both, and 18.8% ($n = 3$) identified with another neurodivergent identity in addition to ASD or ADHD. In terms of diagnostic status, 12.5% ($n = 2$) self-identified without a formal diagnosis, 50.0% ($n = 8$) had received a formal diagnosis, and 25.0% ($n = 4$) were somewhere in between (e.g., they were currently being diagnosed or they had been through initial screening but decided not to pursue further testing)0F.

Although the recruitment information specified that the study was targeted to individuals who identify with traits or diagnoses related to ASD or ADHD, 12.5% ($n = 2$) of interviewees did not explicitly identify with either. Instead, they reported symptoms and traits that aligned with many mental and developmental disorders, including ASD and ADHD. We decided to include these students in our analysis because the approaches and characteristics they mentioned as they explained examples were similar to those of other interviewees.

Throughout the remainder of this paper, instances where we refer to "neurodivergent [engineering] students" reflects participants in the sample, our operational definition (i.e., traits and diagnoses that correspond with ASD or ADHD), and includes those who self-identify as neurodivergent or have a formal diagnosis.

## Materials

Students completed a conflict management style (CMS) assessment prior to participating in semi-structured interviews [57]. The results from the individualized reports generated from ITP Metrics served as a bridge to explore qualitative aspects of teamwork and conflict during interviews. In turn, interview data provided a rich source of information about neurodivergent students' experiences with teamwork and allowed me to understand common threads between individual perceptions and accounts.

For the first part of the interview, McKenna asked students to reflect on their results from their CMS reports and prompted them with follow-up questions to explore whether the results aligned with their self-perception. Then, she asked students about their emotions, cognitions, and reactions as they related to past team conflict experiences. To support a conversational tone and create a safe space for sharing [58], she concluded with a few broader questions about university and course-based experiences and invited students to share any other reflections. The full interview protocol can be found in S1 Appendix.

## Analysis

McKenna conducted the coding of interview transcripts and Thomas O'Neill consulted on the analytic approach and helped structure and organize the codes into themes. The initial phase of data analysis involved cleaning and organizing interview transcripts. By listening to audio recordings to ensure transcript accuracy, McKenna was able to familiarize herself with the topics and tone of each interview. After importing transcripts into NVivo 12, a qualitative analysis software, she conducted a high-level coding process that categorically distinguished between behavioural and cognitive dimensions of responses related to students' teamwork and conflict experiences. These two dimensions align with the study's objective of understanding what neurodivergent students *think* and *do* in team contexts. Furthermore, this structure enhances interpretability and makes connections between themes clear [59].

Given the lack of theories and frameworks regarding neurodivergent experiences within neurodiverse teams, the next phase involved inductively coding the excerpts within the cognitive and behavioural dimensions. By coding line by line, McKenna created over 20 descriptive codes for each dimension, which helped her uncover and understand the meanings that students assign to their teamwork experiences. Throughout this process and in consulting with Thomas, clusters of codes naturally emerged, particularly where certain perceptions or actions were clearly demarcated or where they contrasted (e.g., direct assertion versus hesitant questioning). We created memos to record ideas about the relationships between codes or guidelines for differentiating similar codes [60].

The final phase of the analysis involved developing a coding structure to document the coding relationships and hierarchies. Throughout the descriptive coding process, we had hunches about how codes fit together; however, for more ambiguous codes, we used highlighting to explore various options for grouping codes that seemed to "hang" together. This process was instrumental in defining themes and refining the study's theoretical constructs. Table 2 provides an overview of themes and their definitions. The complete coding framework can be found in S2 Appendix, which shows how we aggregated descriptive codes into themes.

## Findings

Our findings provide insights into the ways in which neurodivergent engineering students make sense of and interact within team settings, including their approaches to managing conflict. The themes and data are presented in cognitive (sensemaking) and behavioural (interacting) dimensions, which highlight what neurodivergent students *think* and *do* as they face and manage challenges and opportunities within team settings. "Sensemaking" refers to how students interpret and understand teamwork interactions and conflict, while "interacting" describes the strategies and actions they employ to manage challenges. In the following sections, we describe these two dimensions and the themes that fall within them in detail.

## Cognitive

The cognitive dimension details what features of teamwork situations are salient to students and how they process them. The findings underscore an awareness of how neurodivergent traits are perceived by peers, as well as the tension between students wanting to express themselves authentically while fearing misinterpretation and judgment from others. Interestingly, students committed time and energy to understand their team members, allowing them to perspective take and demonstrate empathy, as well as develop an awareness of shifts in team dynamics. Given the degree of social

**Table 2. Aggregate Dimensions, Themes, and Theme Definitions.**

| Aggregate Dimensions | Themes | Definition |
|---|---|---|
| **Cognitive (Sensemaking)** | Interpreting identity in team context | How a student perceives and interprets their own identity within the team's identity, encompassing their need for recognition and validation. |
| | Exercising interpersonal sensitivity and empathy | A student's ability to empathize and be cognizant of others' viewpoints and emotional states, as well as the tensions within the team. |
| | Processing internal dialogue | The inner conversations, reflections, and mental conflicts a student experiences when analyzing their own and their team members' approach to working together. |
| | Conflict narrative | A student's perception of conflict and how they frame or interpret its role in team dynamics, growth, or problem-solving. |
| | Processing emotions | A student's reflections on and grappling with the emotions arising from teamwork experiences, whether internally driven (e.g., guilt about contributions) or related to external team dynamics and interactions (e.g., frustration with group members). |
| **Behavioural (Interacting)** | Approaches to participation | How a student engages with the team's work and discussions, from championing a vision or proactively initiating action to encouraging collective involvement or following an existing plan. |
| | Approaches to idea exchange | The manner in which a student expresses their own thoughts and responds to suggestions or concerns raised by team members. |
| | Conflict management strategies | The methods a student employs to handle disagreements or interpersonal friction, ranging from withdrawal and monitoring to mediating and confronting. |
| | Role and responsibility distribution tendencies | The tendency of a student to assign (or not assign) tasks and roles to themselves or others to optimize team functionality and personal comfort within the team structure. |
| | Self-management and regulation | How a student recognizes and deals with personal challenges and regulates their behaviour and emotions in the context of team projects. |
| | Strategies to cultivate team cohesion | Efforts made by a student to build interpersonal connections and create an inclusive, understanding team environment. |

information students were taking in and processing, some reported continuous internal dialogue and reflections, or "over-thinking." Often, this internal dialogue was accompanied by emotions that arose as students reflected on team interactions or emotional exhaustion as a result of continuously trying to make sense of perceived neurotypical norms. Finally, how students framed conflict shaped their willingness to engage.

**3.1.1. Interpreting identity in team context.** Students articulated a deep awareness of how their neurodivergent tendencies and traits were perceived within their teams, with some reflecting on how they undervalue their unique cognitive abilities and contributions. Specifically, students explained how their neurodivergence contrasted, and sometimes complemented, their team members' thinking styles:

> …we don't notice the little things everyone else notices. We notice the differences, right? That's what I found, at least. So, my group members will all notice something, and that's their way to fix it, but in my mind, I notice it from a different perspective. (S9)

Students described a mismatch between how they view their own intentions and behaviour and how their team members interpret them. One student explained that they can come off "as something else" when they are just trying to engage in the conversation and share their knowledge (S12). This discrepancy often led to misunderstandings, with intended cooperative or neutral actions being misconstrued as unresponsive or aggressive, as one student recounted:

> But aggressive, to attack, to go after somebody, that's what shocked me. I genuinely didn't think I was that person, right? I think that was the main piece of it, that I never thought I went for anybody like they're depicting I was. (S9)

Given the evidence that students had from previous teamwork experiences, others regularly misinterpret their actions and behaviours, typically assigning negative intent. Therefore, it is not surprising that these experiences culminated in a broader sense of feeling constantly judged or misunderstood by others and struggling to gain respect, as illustrated by the following student:

> It's like, "Oh. Why would you say that? Why would you think that?" And it just causes a lot of misunderstanding. (S4)

Closely connected to these sentiments is the need to feel acknowledged, validated, and respected by team members, as explained by a student:

> In terms of communication, when I find myself in environments where the way I naturally communicate is seen as a strength, I can be close to people in a way that is meaningful to me. (S11)

Taken together, these reflections represent a constant tension between the desire for authentic expression and the challenges of being truly heard and validated as a neurodivergent individual.

### 3.1.2. Exercising interpersonal sensitivity and empathy.

Empathy and sensitivity toward the feelings and perspectives of team members were brought up frequently. Clearly, the desire to be understood, as outlined in the previous theme, also translates to the desire to understand others, including taking time to understand the overall atmosphere of the team environment, as one student described:

> In general, are people very monotone, quiet, and serious? Are people talking at all? On the other hand, are people very lively—almost to a detriment because it's off-topic because we're having so much fun together? (S14)

Students highlighted both cognitive empathy (being able to see things from another team member's perspective) and emotional empathy (personally connecting with and relating to the feelings of other team members). For example, one student explained that they can both put themselves "in the shoes of other team members" in addition to being able to "feel when another team member feels left out" (S5). Considering another team member's perspective was not always second nature for students, especially in contexts where emotions were heightened. One student shared, "If I'm part of the conflict, then it might take me a while to see it from the other person's perspective. But if I'm not part of the conflict, I'll try to see it from both sides" (S8).

Subtle changes, such as a team member talking less or seeming distant, were signals students noticed that could indicate distress. One student described, "I was like, 'I'll do the whole speech,' and then one of my friends insisted that he wanted to do it, but when he went up, he kind of choked… And I just felt bad for him" (S4). Attentiveness created a foundation for students to behave in a more accommodating way, taking team members' emotions, life circumstances, and commitments into account, as illustrated by a student who expressed, "I get it if something happens. I'm not a 'No, too bad, you have to do this' type of person" (S10).

Closely related to empathy, students reported being highly attuned to tensions within the group and how they might affect fellow team members, which sometimes impedes their ability to work effectively. One student described, "Say there's someone with a very dominating presence in the group and they're excluding someone, I definitely get overly empathetic about it, where I'm unable to work until it's resolved" (S16).

Students' ability to empathize and be cognizant of others' viewpoints permeated many interactions and revealed the dual desire of wanting to connect with and understand others while feeling understood themselves.

**3.1.3. Processing internal dialogue.** Internal dialogue represents the internal thought processes and negotiations that occur when students evaluate team dynamics or decide how to address team conflict and express their ideas. Students described a constant internal conversation that influences their decision-making and interaction strategies within teams. For example, one student explained how overthinking can lead to difficulty taking action by stating, "I overthink a lot in my head, and I communicate a lot with myself in my head, but I can't always communicate with others" (S5).

This pattern of hyper-fixating on and thinking deeply about one thing at a time can result in a rigid approach to evaluating people, ideas, and situations, which can make it difficult to reconsider initial judgments. One student shared, "I definitely struggle to go from disliking someone to liking someone. It's either 'yes' or 'no,' and I tend not to go back and forth too much" (S6).

Students carefully weigh the potential risks (e.g., conflict, judgment) and benefits (e.g., clarity, contribution) when deciding whether to voice their opinions and speak up in team interactions:

I feel inside, like I don't like the direction we're going in, but I feel like it's still just easier because I don't have a clear, distinct idea in my head of where we should be going. (S1)

Together, these reflections underscore how constant internal dialogue can, in many instances, result in feeling "stuck," whether due to hesitation in sharing opinions or difficulty shifting initial evaluations.

**3.1.4. Processing emotions.** Students reflected on their experiences grappling with the often-complex emotions that arise during teamwork. These emotions could be internally driven (e.g., guilt about contributions) or related to external factors and interactions (e.g., frustration with team members). Although team members were unaffected by instances where they internalized frustration or negative emotions, students experienced emotional tension that impacted their perception of their team members, often negatively. As one student highlighted, there are emotional consequences to constantly internalizing:

I don't really have conflict with people, but something that does end up happening, though, is I build up emotions towards people. Sometimes I'll get frustrated but I won't express it. Then, when I look at the person, I'll just get frustrated for no reason, just because I'm looking at them, and I associate them with frustration. (S5)

Uncertainty also played a significant role in triggering negative emotions. One student likened the experience of having little control over outcomes to "being on rails, like a train… You're just going where the rails are going" (S14). Another student described the paralyzing effect of uncertainty and how it can leave them feeling hopeless and anxious, which drives their direct approach to addressing conflict:

It makes me very anxious whenever I've gotten messages from team members saying, "Hey! We have an issue." And then that's it… it actually will be like 50% of what I'm thinking about for the rest of the day. So, then I'm like, "Okay, we need to solve this now." (S10)

Students recounted the considerable emotional labour involved in conforming to perceived neurotypical norms, including interpreting implied meanings and navigating communication expectations, as illustrated by a student who remarked that they could not "constantly be guessing at what they [their group members] mean" for a semester-long project (S11). Another student explained, "I don't think I always know how to talk to certain kinds of people because there seem to be different styles of communicating for different types of people. Generally, I use the same style with everyone I talk to, and it doesn't always work" (S16), highlighting how relying on their natural approach to communication can lead to misinterpretations and tension.

One student described the strain and exhaustion they experienced as a result of masking:

I didn't realize how much energy I was focusing into masking and having to "act normal" until my counsellor gave me back my scores on it, and I apparently do that a lot, so that also factors in. And energy and burnout—just having to consciously make the decisions to act more like a neurotypical, just like what's expected of me, you know? (S12)

**3.1.5. Conflict narrative.** How students interpret conflict (e.g., positively or negatively valenced) affects how they react to and engage in difficult conversations. One student discussed how they view team conflict as a shared challenge:

To me, it's not us against each other; it's us versus a problem. So why don't we work together to solve the problem instead of fighting each other over it and avoiding it? It just prolongs how long someone's mad at someone else. (S10)

Conversely, other students held a belief that conflict is inherently negative, destructive, or damaging to team relationships:

I don't want to be in anyone's bad eye, so I won't really say anything. And again, I tend to promote safe work environments and safe classroom environments. So, I feel like if I did do that [confronting a team member about their lack of involvement in the project], I would be affecting what I'm trying to create. (S15)

A tension between the desire for immediate resolution and the need for delayed or more reflective problem-solving was apparent. One participant described the pressure they felt to quickly settle disagreements by stating, "…it also means sometimes that problem needs to be avoided for now, and we need to put a little circle around it and say we're coming back to this. I have a tough time doing that. I'm really like, 'Let's solve this. Let's do it'" (S10). This balance between taking immediate action and allowing time for reflection represented an ongoing negotiation between the emotional urgency of conflict and the desire for a comprehensive solution that represents the team's collective interests and perspectives.

**Behavioural**

The behavioural dimension details specific actions or strategies students often employ or have employed in the past in teamwork situations. In participating in team discussions, students' strategies ranged from asserting ideas and confidently challenging team members to stepping back and allowing other team members to take the lead. The conflict management approaches students gravitated to aligned with and, in some cases, departed from the five styles in the CMS assessment. *Uncertainty* was a common thread between these themes. In some cases, students' discomfort with ambiguity and uncertainty about their standing with team members prompted them to adopt more direct approaches. However, in other cases, this uncertainty, coupled with the desire to maintain team harmony, led students to adopt more cautious approaches. This desire to reduce uncertainty may also have motivated students to shift their roles to fill a void or respond to situational demands as the team's work evolved. Students acknowledged executive functioning challenges when carrying out tasks, but leveraging team support allowed them to stay on track. Finally, by modelling open communication and acknowledging team members' opinions and contributions, students promoted an inclusive team environment.

**3.2.1. Approaches to participation.** When discussing how they participate in projects, students highlighted how they typically contribute to the team's work and discussions. These contributions ranged from students generating and pushing ideas to following directions and stepping back to allow others to take the reins. Many students sought to create opportunities for *all* their fellow team members to voice their opinions and ideas:

And so, I feel like the biggest success is just making a product that everyone involved in and can be proud of it. Everyone can say, "Oh, I did this part. I did this part." As opposed to everyone being like, "Oh yeah, I did this because you know, so and so told me to." (S7)

On one end of the continuum of influence and compliance, some students gravitated toward actively, and sometimes assertively, shaping the team's direction. One student illustrated this approach by sharing, "…but I'm not good at being like, 'Oh, we can do your thing.' I'm not good [laughs] at that. I'm pretty stubborn, and I can be insulting accidentally" (S6). In contrast, on the other end of the continuum, students prioritized following direction from their team members as opposed to asserting their own ideas. One student shared, "So I kind of feel like I shouldn't be trying to input into the initial idea of a project. Because I can follow through with a process. That should be more of my contribution to the project" (S1).

In addition to the degree of influence students preferred to have in projects, their orientation to taking action also emerged as a key approach. Some students described how they consistently advocate for addressing potential obstacles as soon as possible and prioritizing progress. One student remarked, "I don't avoid problems. If there is an issue, it is a 'let's solve it now issue' all the time" (S10).

**3.2.2. Approaches to idea exchange.** Students employed a range of strategies when conveying and responding to ideas and concerns during group discussions. Some students described confidently challenging, debating, or pushing others to think more deeply. This direct assertion was illustrated by a student who explained, "As soon as I hear someone's idea, I automatically play devil's advocate. I just jump right to, 'What about this? Have we thought about how we would do that?'" (S5). Other students gravitated to a more cautious approach to speaking up by using "softer" language and qualifying statements. One student shared, "When I do speak up about something—which I do—it's more like, 'Maybe we should rethink this part' not, 'I think we should do this instead'" (S1). Another student described, "If I'm ever in a group setting where I don't exactly know everyone, then I'm a bit less comfortable, especially when I have to share my idea" (S13), highlighting the interplay between hesitance in speaking up and a sense of psychological safety in the team environment.

Some students detailed instances where they explicitly checked in with team members to confirm agreement or ensure collective buy-in on proposed actions or directions. Rather than assuming everyone was on the same page, they asked questions such as, "Does that sound okay?" or "Does anyone have any objections to that?" (S14).

Even though students take different approaches to sharing ideas during discussions, they carefully assess the potential impact of *how* their questions and opinions are phrased and how this might influence group dynamics and project outcomes.

**3.2.3. Conflict management strategies.** Although students' methods for managing team disagreements varied, they demonstrated awareness and attunement to situational cues, especially team members' emotions. Some of these strategies closely aligned with the five styles students would have received scores on in their CMS report, while others were unique or represented more nuance in how a certain style was enacted.

For instance, many students highlighted a tendency to gravitate to a mediator role for other team members. One student shared, "If I'm seeing conflict in my team, I'm able to be like, 'Okay, let's talk it out. What's going on? What are we feeling this way?'" (S5). Although this reflection represents aspects of the *integrating* style, it also demonstrates a deeper focus on facilitating open dialogue and supporting team members' emotional well-being.

Examples that highlighted withdrawal from or monitoring of conflict mapped well to the *avoiding* style. For example, one student explained, "…if there was anything wrong in a group environment, I'm like, 'Okay. I don't want to talk about it. I don't want to do this'" (S15). However, instances of a student observing group dynamics and noticing tensions, while avoiding direct confrontation, go beyond the definition of *avoiding* because of the focus on gathering information. This attentiveness is illustrated by a student who remarked that they typically "spectate conflict" (S12), suggesting that taking a step back from conflict could be a way to enhance understanding of the situation rather than purely to escape the discomfort associated with confrontation.

Opposite to the *avoiding* style, some students preferred to address conflict directly. One student recounted a situation where they confronted a team member about their lack of involvement:

And so, we were kind of like giving him crap for it. Like, "Come on, man. We feel like you're not pulling your weight on this group project, and then we're all doing more than you." (S7)

While the focus of the *dominating* style is on persuading others to accept one's own preferences and goals, the aim of direct confrontation is to clarify expectations and promote collective accountability.

However, some students worked in situations where there was a lack of clarity and parameters. Navigating this grey area pulls in elements of *avoiding* and *accommodating*. Students sacrificed their desire for a clear path forward to maintain group cohesion and relationships. One student illustrated their willingness to work in ambiguity to maintain peace:

The problem is when we don't have an idea of where we should go. Some people may have some vague ideas, and I might still try to work with it. Like, "Okay, well, you have this idea. Let's try to put this together." But [the issue is that] it doesn't fit… (S1)

In a similar vein, students described instances where they conceded their needs, opinions, or preferences for the benefit of the group or a specific team member, which mapped closely to the *accommodating* style. One student recounted, "Rather than discussing it like I had first project, I just said, 'Okay, we can work with yours' in that attempt to be more cooperative" (S9). One student described withholding their contributions as a form of protest, which did not align with the five styles:

…they can struggle because they actually don't know the things I know about this. So, I ignored all the conversations about it and anything, and I didn't offer any expertise that I had. (S9)

This strategy may have provided an avenue for re-establishing some degree of autonomy and control that this individual felt had been taken away from them.

**3.2.4. Role and responsibility distribution tendencies.** In discussing how they navigate their roles within teams, students reflected on their tendencies to assign roles and tasks to themselves or others to optimize team functioning. In terms of roles, concepts of flexibility and (lack of) clarity emerged from the data. In explaining how they "found" their role, one student recounted, "Pretty quickly in I realized that like everyone was like kind of picking roles and the leader role had kind of already been taken, and I wasn't really sure like how I would work in that role so I definitely didn't even try to be that person" (S6).

Furthermore, students adapted their roles when they noticed a leadership void, a situation demanded a certain skillset, or specific responsibilities needed to be filled to meet the team's needs:

Whatever role is missing, I fill that role. Like in my family role, it's the quiet, compromising, just mediating role. At school, it is a delegation kind of leader, more outspoken role. (S9)

Interestingly, despite students' discomfort with uncertainty, these reflections demonstrate that many were willing to work in tasks and roles without clear parameters to establish direction and focus. Beyond being adaptable, students distributed tasks to ensure equity and effectiveness, as well as to reduce uncertainty and personal anxiety. Some students delegated tasks with consideration of team members' abilities, commitments or personal circumstances, such as one student who stated, "I like to make sure that if people have other things going on, besides our project, that there's like a reasonable expectation set for them" (S16). Another student described how they assign tasks by explaining, "…it's just divvied up by 'you do this, I'll do that. Does that sound fair?'" (S2).

Other students preferred not to delegate and instead handle tasks themselves. In some cases, this preference was driven by the need for efficiency under time constraints, as highlighted by one student who said, "When they left the room, I was very able to be super efficient and get everything done" (S6).

**3.2.5. Self-management and regulation.** Students discussed how they handle the pressures and demands of teamwork while regulating their behaviour and emotions. Some students experienced challenges with tasks and responsibilities that hinge on executive functioning abilities (e.g., organization, time management), as exemplified by one student who shared, "…I have a million ideas going on, but I don't ever finish them, so I need one of my friends to write them down as I go, so I can revisit them because I will forget and just move on to something as I get a thought" (S3).

Although students acknowledged how executive functioning challenges related to their neurodivergent identity could negatively impact their teamwork contributions, they also leveraged team support to overcome these obstacles. One student alluded to the benefits of "body doubling" by sharing, "…as a person who struggles with ADHD, I feel like being in a team really just balances out all the downsides of my ADHD, you know. Other people might keep me in control and stop me from getting distracted" (S4).

Lastly, students modelled transparent communication in sharing project-related updates, acknowledging issues, and inviting frank discussions to maintain clarity and understanding within the team. One student explained that if they experienced some kind of challenge and were delayed in their contributions, they would go to their team members and say, "This is what's going on my end, and I will be fixing it at some point" (S3). In most cases, this transparency was linked to both wanting to contribute meaningfully to the team's work and minimizing instances of ambiguity.

Together, these reflections reveal how students draw on supportive structures, transparent communication, and self-awareness to navigate challenges related to, and that extend beyond, executive functioning.

**3.2.6. Strategies to cultivate team cohesion.** Lastly, students emphasized the importance of fostering an inclusive team culture where team members' identities are respected and valued. Building a foundation of trust and rapport was often prioritized, as exemplified by one student who noted, "…having a stronger relationship with my team members would, of course, be the first thing. It just makes dialogue a lot more easy and makes communication a lot more easy" (S5).

Team connection and cohesion hinge on recognizing and valuing the diverse abilities of all team members, which, tying back to the cognitive dimension, directly impacts how individuals perceive their own contributions and worth within the group. In some instances, students chose to disclose their neurodivergent identity to their team members to reduce the risk of misunderstanding and assumptions:

And my identity didn't come up in the words. I think after the [references specific situation], I did make a joke of like, 'tism. And then we laughed and moved on. But in that specific conversation, no, it didn't come up in like a, "Yeah, I got screened for XYZ and here's where we are." But it did sort of come up in the way that was like, "Hey, this is how my brain works…" (S11)

Students disclosed to promote understanding, and this understanding extended to team members as well. Valuing the perspectives, opinions, and contributions of their team members were ways in which students promoted mutual recognition and respect. One student reflected, "The thing about me and other neurodivergent students is that we don't judge. We accept people for who they are…" (S15). Similarly, another student stated, "I find other people who have different traits are not always accepted. I don't usually have any bias against that" (S16).

## Discussion

The aim of this phenomenological study was to address how neurodivergent students make sense of and interact within team settings, including their approaches to managing conflict. Given the anticipated rise in diagnoses and identification

within the neurodivergence umbrella [61], it is becoming increasingly important for researchers to address the scant information and evidence on neurodiverse teamwork. Educational and professional institutions can expect a greater proportion of neurodiverse teams as a result of this "rise" in neurodivergence, making this topic timely and practical. An exploration of neurodivergent teamwork experiences is necessary to inform practical frameworks that promote effective neurodiverse teamwork and neuro-inclusivity. Ultimately, integrating these practical frameworks into institutional practices, such as hiring, curriculum design, and policy, better equips organizations and academic institutions to embrace the unique talents and perspectives of neurodivergent individuals.

This study contributes to the body of literature focused on understanding neurodivergent teamwork experiences through the use of semi-structured interviews. Over 50 codes emerged from the interview data, which we organized into 11 themes that represent cognitive and behavioural dimensions of teamwork experiences. Together, these themes offer insights into how neurodivergent engineering students experience teamwork and conflict through cognitive and behavioural dimensions. The following sections are illustrative of our sample, in which students self-identified their neurodivergent status. Therefore, the patterns we outline and interpret do not represent the teamwork experiences of all neurodivergent individuals or those with a specific clinical diagnosis.

## Interpretive summary and theoretical contributions

Through conducting this study, we sought to merge two distinct bodies of literature that have evolved independently of each other: conflict management and neurodiversity. On one hand, conflict management theories, embedded within the teamwork literature, have focused on conflict styles, states, processes, and expressions without explicitly accounting for neurodivergence. On the other hand, neurodiversity theories, often situated within EDI and disability literature, have focused on stigma, identity management, discrimination, and barriers to participation with little consideration of the teamwork context. The current study contributes to the integration of these bodies by revealing how neurodivergent students' cognitions and behaviours related to teamwork and conflict are shaped by sensitivity to interpersonal cues, perceived neurotypical norms, and efforts to avoid being misunderstood.

**4.1.1. Conflict management theories.** Students displayed a range of conflict management strategies that both aligned with and departed from the dual concern model. While the *integrating* style is typically characterized by active engagement in the conflict (i.e., individuals advocate for *their* interests and those of the opposing party), some neurodivergent students reported positioning themselves as neutral facilitators, focusing on understanding and mediating interactions *for* their fellow team members. This tendency to assume a mediator role may be related to students' proficiency in perspective-taking and sensitivity to team members' emotional states. Additionally, some students described monitoring behaviours, which may be related to the *avoiding* style, as students reported observing conflict without intervening. However, this behaviour was not merely about evasion; students were observing and gathering information about their team members to better understand the dynamics at play.

Neurodivergent students described heightened sensitivity to team and conflict dynamics. This heightened sensitivity may lead students to adopt strategies to carefully gauge and act on interpersonal cues and emotional tensions within their teams. In some cases, this heightened sensitivity gives rise to unique responses that deepen our understanding of styles in the dual concern model, particularly as they relate to concern for others. For instance, the data illuminates *why* students might adopt the *avoiding* style, which has been a longstanding interest to researchers due to uncertainty around the underlying cognitive and emotional motives for disengagement [62]. What might be interpreted by team members as "aloofness" actually reflects an intentional effort to absorb and process value-laden social information and cues from team members [9]. The way in which neurodivergent students interpret these interpersonal cues to inform strategies may be further explained by social information processing theory..

Departing from the dual concern model, one student described withholding their contributions as a form of protest. Specifically, they withheld their expertise and knowledge from the team to "even the playing field" and re-establish control.

This approach highlights that the dimensions of concern for self and others captured in the dual concern model may be affected by perceptions of equity or threats to autonomy. Therefore, integrating relational justice theories into conflict management style theories may be especially relevant for understanding conflict in neurodiverse teams.

Students demonstrated intentionality and flexibility in enacting conflict management styles depending on team dynamics and situational cues. Many students explained strategically adjusting their approach based on team members' emotional states or the need to promote collective accountability and cohesion. This adaptability in conflict management strategies reflected a broader pattern of intentional flexibility that extended across multiple teamwork behaviours. For instance, in describing their teamwork roles and responsibilities, students explained readily shifting their roles in response to perceived leadership voids and project demands for a specific skillset, highlighting a broader ability to adaptively engage with teamwork challenges. Additionally, when participating and exchanging ideas, students strategically took active roles in guiding discussion or driving project progress while also recognizing when to step back to give other members the chance to engage and influence the team's collective direction.

Research on conflict management styles recognizes that there is *no single best method* for handling conflict across situations and that methods *should* vary across contexts [39,41]. In line with these propositions, the findings offer some preliminary insights into *how* and *why* neurodivergent students shift among behaviours and strategies. This fluidity reflects a process-based perspective of conflict management, in which styles are not a one-time choice, but iterative, situation-to-situation decisions that accumulate to influence broader teamwork patterns [38,40]. Furthermore, the findings position conflict management within the broader social dynamics of teamwork. Rather than treating approaches to managing conflict in isolation, the data reveal that neurodivergent students calibrate their actions to maintain trust, cohesion, and accountability across multiple teamwork domains.

**4.1.2. EDI and neurodiversity theories.** An overarching theme that emerged from the data is the dual desire to be understood and understand others. Many students described being acutely aware of how their neurodivergent traits might be interpreted by team members, sometimes anticipating judgment. Simultaneously, students invested significant mental effort in observing team members, anticipating others' needs or emotional states, and reading the overall team dynamic. These cognitions laid a framework for several corresponding behaviours, such as confirming agreement among team members, using qualifying language to soften suggestions, or, conversely, speaking up if they felt a team member was being excluded. Concern for their team members and the need for their own neurodivergence to be acknowledged compelled students to foster team environments where mutual understanding and inclusion were central tenets.

These cognitive and behavioural efforts align with the social model of disability, which underscores the role that institutions and peers play in creating barriers for neurodivergent individuals to participate [17,63]. Some students noted the effort they exerted in piecing together social information in interactions that seemed readily available to their neurotypical team members. Structural stigma also drives individuals to engage in identity management behaviours to reduce their risk of experiencing discrimination [20,64]. One student, in particular, referred to "masking" as a way to make their neurodivergent characteristics less noticeable in team contexts. Stigma awareness and anticipating judgment led to instances where students had to reconcile the differences between intended and perceived messages or grappled with feelings of inferiority about the value of their abilities, knowledge, and ideas. These findings suggest that the stigmatization of neurodivergent traits creates barriers to participation and inclusion and can negatively impact self-identity, which aligns with other research [52].

Collectively, these findings highlight how institutional norms and peer expectations impact neurodivergent students' concept of their own identity, which in turn influences their behavioural strategies and approaches. Although relationships between institutional norms, stigmatization, discrimination, and how neurodivergent individuals perceive themselves have been previously established in the neurodiversity and EDI literature, this knowledge has yet to be applied to neurodivergent teamwork experiences.

Despite these points of clear connection, the findings suggest that some theories attempting to explain the social elements of neurodivergence are limited in their capacity to capture the nuanced ways in which neurodivergent individuals

seek to understand and relate to others. Specifically, the double empathy problem, coined by Milton [48], describes how "different dispositional outlooks and personal conceptual understandings" can increase the potential for bi-directional misunderstanding between neurodivergent and neurotypical individuals. However, the themes we derived from interviews suggest that neurodivergent students are extremely sensitive to their team members' emotional and cognitive states, often going to great lengths to ensure all team members feel valued. This heightened awareness may result in neurodivergent individuals taking on significant emotional labour within the team as they navigate their neurodivergent identity and teamwork dynamics.

The "double" empathy problem may be more reflective of a lack of reciprocal understanding from neurotypical individuals towards their neurodivergent peers. Based on our interactions with neurodivergent students, we found that they actively work to bridge gaps in understanding and communication. Future research is needed to explore how neurodivergent and neurotypical team members perceive and reciprocate efforts to bridge differences. Representational gaps (rGaps) offer a broader lens for understanding underlying processes that drive miscommunication [49]. In contrast to the emphasis on empathic shortcomings in the double empathy problem, rGaps extend these fundamental differences to underlying representational systems used to interpret information. Ultimately, rGaps may offer a more robust framework for identifying, reconciling, and leveraging differences in representational systems to bridge gaps between those with different perspectives and lived experiences.

## Practical implications

Concrete criteria and frameworks can help educators improve the teamwork experiences of neurodivergent students and promote effective neurodiverse teamwork. During the conclusion of interviews, many students shared actionable insights for how teamwork and conflict could be better supported in postsecondary settings. Table 3 presents recommendations for course designers, instructors, teaching assistants, and students based on interviewees' experiences and suggestions. For example, course designers might consider integrating opportunities for team members to connect, share personal experiences, and understand each other's working styles to support teams in building a foundation of trust and psychological safety. The CMS assessment used in this study can help students get to know each other quickly by encouraging them to reflect on their tendencies and discuss how their styles might influence team dynamics [57]. Additionally, instructors might consider presenting evidence-based frameworks to help teams communicate, make decisions, and mitigate negative conflict. The SUIT framework breaks down the principles that underlie constructive controversy theory into easy-to-remember steps and has demonstrated ability to improve team interactions in student learning teams [33,65]. Taken together, the recommendations presented in Table 3 may offer educators tangible starting points and serve as a foundation for future researchers to develop robust practices and frameworks for neurodiverse teamwork.

## Limitations and future research directions

While this study sheds light on neurodivergent students' teamwork experiences, there are a number of limitations. First, we used a narrow definition of neurodivergence for recruitment (i.e., self-identified traits and diagnoses that align with ASD and ADHD), and therefore, our sample did not represent the full spectrum of neurodivergent identities, traits, and experiences that exist across post-secondary populations. We justified our inclusion criteria with evidence to suggest similar cognitive patterns underlie ASD and ADHD [66] and to interpret findings meaningfully within the existing literature (much of which seems to be oriented to ASD and ADHD). However, inclusion criteria seem to differ drastically among previously conducted studies on and with neurodivergent populations. Future researchers should consider expanding their definition and inclusion criteria for neurodivergence (if appropriate for the study design) to explore less commonly discussed identities and traits (e.g., specific learning disorders such as dyslexia, communication disorders such as speech sound disorder). Moreover, in a similar vein to the practical implications we outlined, it would be valuable to investigate how diverse cognitive traits and thinking patterns influence team interactions and dynamics. Second, only 16 students'

**Table 3. Recommendations for Educators to Support Effective, Inclusive Neurodiverse Teamwork.**

| Educator Role | Team Functioning | Conflict Management | Evidence |
|---|---|---|---|
| **Course Designers** | • Integrate a component into the course syllabus that clearly outlines baseline expectations for teamwork engagement and contributions<br>• Embed structured activities for teammates to learn about each other at the beginning of each group project (e.g., Personality or Conflict Management Styles assessment through ITP Metrics)<br>• Introduce shorter, low-stakes projects before the high-stakes term project to allow students to build confidence and take on new roles | • Incorporate team contracts that have sections to document expectations and norms that are relevant to student teams (e.g., meeting management, accountability measures, etc.)<br>• Develop specific and actionable accountability mechanisms (e.g., formal action plan, grade deduction) and avenues for expressing concerns and feedback (e.g., office hours, anonymous Google form) for students who do not meet teamwork expectations outlined in the syllabus | "So giving a week or two just to meet the people, understand how comfortable they are in different situations, how much they like to talk, what their typical behaviour is, if they're really pushy, if they're not pushy, if you can become close friends with them, how you're allowed to talk to them" (S13).<br>"My courses require us to do what we call team contracts… so I think that's helped me learn to speak up and not just take things" (S12).<br>"Some good examples of this I've seen are anonymous feedback or just being able to have a feedback mechanism in place, so I know that I'm allowed to give this feedback" (S5). |
| **Course Instructors** | • Frame diversity (including neuro-diversity) as being beneficial to teamwork (i.e., diverse perspectives can fuel innovation)<br>• Outside of structured icebreakers, encourage ongoing dialogue about working styles, feedback preferences, and triggers<br>• Model explicit communication in outlining the teamwork expectations for the course, as well as providing written and verbal instructions for assignments | • Intervene according to the teamwork expectations and accountability mechanisms outlined in the course syllabus<br>• Integrate course-wide teamwork check-ins and training that aligns with intense working periods leading up to deliverables (conflict often arises during these periods)<br>• Provide guidelines and frameworks for managing conflict constructively (e.g., "SUIT") | "The thing I struggle with in conflicts is just not understanding what's being implied or what I have to get" (S12).<br>"I wonder if it would help spending more time teaching teamwork skills in first year" (S16).<br>"So, maybe more *considered* questions to get the ball rolling when it comes to just talking with your group members" (S2). |
| **Teaching Assistants** | • Normalize different approaches to the task, assignment, or project at hand (e.g., visual vs. analytical reasoning)<br>• Model direct, yet respectful ways to ask questions and give feedback during informal check-ins or in written feedback for assignments | • Proactively check in with teams so students who are hesitant to approach instructors can still voice concerns.<br>• Empower and guide teams to problem solve independently before escalating to the course instructor (e.g., "GROW" coaching model) | "Even me talking to the prof is sometimes kind of intimidating. So I feel like if the prof would come by and actually take a look at my group and see what's happening, then it'll kind of be a little better because I know the prof is aware of the situation" (S15).<br>"So they just try to say 'this [tip] is helpful,' as though it's a fact, and it might not be. Just the understanding of that" (S1). |
| **Neurodivergent Teammates** | • Communicate needs and working preferences<br>• Develop strategies for self-advocacy and boundary setting | • Acknowledge when challenges related to neurodivergent status affect contributions, and communicate proactively | "I think it would help neurodiverse people to let their group mates know what their triggers or 'set offs' are" (S9). |
| **Neurotypical Teammates** | • Demonstrate patience, flexibility, and openness to diverse working styles<br>• Foster an inclusive team environment where input and feedback are regularly invited | • Demonstrate willingness to learn about challenges neurodivergent team members experience, and support them in problem-solving if appropriate | "When it [individual contribution] didn't work out, they still trusted me to figure it out *with* them, and also as my own person to figure out a solution for that" (S3).<br>"People not taking offence to things easily because I ask a lot of questions. And if you ask too many questions, people can get pretty irritated and whatnot" (S6). |

S = student. The number following the "S" indicates a specific interviewee.

perspectives were analyzed and represented despite our recruitment efforts in multiple engineering courses. Although this is deemed appropriate for a phenomenological study, it is in the low-to-mid range of recommended participant numbers [67]. Excerpts from three or more students' interviews were represented in most of our descriptive codes; however, there were a few codes that were unique to one or two students. We decided to keep these codes as they were of theoretical significance. More data would allow for a comprehensive understanding of the prevalence of each theme and capture a greater breadth of experiences. Future researchers should aim to recruit larger and more diverse samples to extend and refine the themes and framework presented in this study. For example, researchers might adapt the semi-structured interview guide for text-based responses, allowing them to quickly and easily gather data from multiple institutions, organizations, and community groups.

Third, findings cannot be generalized beyond boundary conditions and contexts that are similar to this study. Students self-reported whether they identify as neurodivergent, which resulted in a sample that was heterogenous with respect to diagnostic status (i.e., only half of students had received a formal diagnosis). Accordingly, our findings are not generalizable to specific clinically defined psychological conditions, as our sample was not representative of diagnostically homogenous ASD or ADHD populations. Beyond diagnostic considerations, further exploration is necessary to determine whether findings from this study would extend to professional settings. For instance, compared to students, professionals would likely have years of teamwork experience, work in teams with a longer tenure, and be motivated through different goals and incentives (e.g., compensation, job security, promotions). Corbin and Strauss [68] suggest that after selecting a homogenous group of individuals who have experienced a phenomenon, theoretical sampling might entail studying a heterogenous sample to further understand the conditions in which the initial framework or typology holds. Applying this approach to the current study might mean interviewing students in other programs and professionals at varying stages in their careers about their teamwork experiences.

Fourth, given the phenomenological nature of this study, there was no neurotypical comparison group. Consistent with theoretical sampling principles, participants were deliberately selected based on their potential to inform and refine our understanding of neurodivergent teamwork experiences (i.e., all participants identified as neurodivergent and had recent group project experience). Accordingly, we cannot determine whether the patterns outlined and interpreted are unique to students who self-identify as neurodivergent or reflect broader teamwork experiences among undergraduates. Future researchers should employ comparative designs to examine similarities and differences across neurotypes.

While developing practical frameworks for educators and policymakers to support neuro-inclusive team environments is a logical next step from the current study, a "nothing about us without us" approach should be adopted. Researchers should collaborate with neurodivergent individuals to develop frameworks that offer meaningful, lived-experience-informed support for neurodiverse teams. Collaboration is especially important given the underrepresentation and devaluation of neurodivergent individuals in research and social institutions [17]. Participatory approaches validate the experiences of neurodivergent individuals and contribute to a more in-depth understanding of the social world where individuals of all neurotypes interact [69,70].

## Conclusion

The current study explored the teamwork experiences of neurodivergent students by analyzing how they made sense of and interacted during instances of team conflict. 11 themes emerged from the data, which we categorized into cognitive and behavioural dimensions. In synthesizing connections between these themes, we outlined how our findings relate to conflict management theories and neurodiversity theories, underscoring both the need and opportunity to further integrate these relatively independent research bodies. Furthermore, we presented actionable recommendations for educators to support course-based teamwork by drawing on the experiences, challenges, and suggestions students shared during interviews (Table 3). Ultimately, this study provides a starting point for understanding neurodiverse teamwork, which is essential for informing comprehensive frameworks that promote team effectiveness and inclusivity.

## Supporting information

**S1 Appendix. Semi-structured interview questions.**
(DOCX)

**S2 Appendix. Complete coding framework.**
(DOCX)

## Author contributions

**Conceptualization:** McKenna Sperry, Thomas A. O#39;Neill.

**Data curation:** McKenna Sperry.

**Formal analysis:** McKenna Sperry.

**Investigation:** McKenna Sperry, Thomas A. O#39;Neill.

**Methodology:** McKenna Sperry, Thomas A. O#39;Neill.

**Project administration:** McKenna Sperry.

**Resources:** McKenna Sperry.

**Supervision:** Thomas A. O#39;Neill.

**Visualization:** McKenna Sperry.

**Writing – original draft:** McKenna Sperry.

**Writing – review & editing:** McKenna Sperry, Thomas A. O#39;Neill.

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
