## [Decision Letter · Decision Letter 0]

5 Jan 2026

PONE-D-25-53622Understanding Teamwork Experiences of Neurodivergent Students: A Phenomenological Exploration of Conflict and Collaboration in Engineering TeamsPLOS One

Dear Dr. Sperry,

Thank you for submitting your manuscript to PLOS ONE. After careful consideration, we feel that it has merit but does not fully meet PLOS ONE’s publication criteria as it currently stands. Therefore, we invite you to submit a revised version of the manuscript that addresses the points raised during the review process.

After evaluating the manuscript in light of the reviewer feedback and PLOS ONE publication criteria, I find that the study meets the journal’s standards for technical soundness, ethical conduct, and transparency of data availability. The qualitative design is appropriate for the research questions posed, and the analytic approach is sufficiently described to support the reported findings.

Changes required for acceptance:

The authors must strengthen methodological clarity regarding participant classification. Specifically, the criteria used to define the study population should be more precisely articulated, and the scope of interpretation should be aligned with those criteria. Any statements in the Results or Discussion that could be interpreted as broadly representative of clinically defined groups should be revised to reflect the actual composition of the sample. In addition, the Discussion should more clearly delineate the interpretive boundaries of the findings, particularly where broader group-based distinctions might otherwise be inferred.

Recommended changes:

To further improve clarity and reader guidance, the authors are encouraged to expand the limitations section to explicitly situate the findings within the chosen study design and sampling framework. Minor refinements to the discussion linking empirical observations to the conceptual framework would also enhance coherence, provided these links remain grounded in the data.

Resolution of reviewer advice:

There are no conflicting recommendations among the reviews. All feedback is complementary and can be addressed through clarification, reframing, and strengthened discussion, without the need for additional data collection or changes to the core methodology.

Based on the above, and consistent with PLOS ONE’s emphasis on methodological rigor and transparency rather than perceived impact, I recommend Minor Revision, with acceptance contingent upon satisfactory revision addressing the required points outlined above.

We look forward to receiving your revised manuscript.

Kind regards,

Ramandeep Kaur

Academic Editor

PLOS One

Journal Requirements:

2. In the online submission form you indicate that your data is not available for proprietary reasons and have provided a contact point for accessing this data. Please note that your current contact point is a co-author on this manuscript. According to our Data Policy, the contact point must not be an author on the manuscript and must be an institutional contact, ideally not an individual. Please revise your data statement to a non-author institutional point of contact, such as a data access or ethics committee, and send this to us via return email. Please also include contact information for the third-party organization, and please include the full citation of where the data can be found.

Additional Editor Comments:

Thank you for a thoughtful and timely qualitative exploration of teamwork experiences among neurodivergent engineering students. The manuscript is clearly written, methodologically coherent, and addresses an important gap in higher education research.

To strengthen the rigor and interpretive clarity of the study, the following revisions are required:

Required Revisions (for acceptance)

Clarify Participant Characterization

Please provide a stronger methodological rationale for including both formally diagnosed and self-identified neurodivergent participants.

Clearly distinguish these groups in the Methods section and ensure that conclusions do not assume diagnostic homogeneity.

Reframe interpretations where necessary to reflect perceived neurodivergence rather than clinically defined categories.

Temper Comparative Interpretations

In the absence of a neurotypical comparison group, please avoid language that implies contrastive or causal claims about experiences being unique to neurodivergent students.

Explicitly acknowledge this limitation in the Discussion and refine conclusions accordingly.

Recommended Revisions (to strengthen the manuscript)

Strengthen the Limitations Section

Expand the discussion on how self-identification and lack of comparison groups may influence transferability and interpretation.

Briefly suggest how future studies could address these gaps (e.g., comparative or mixed-methods designs).

Align the Conflict Management Framework

Clarify whether the conflict framework is used descriptively or interpretively, and ensure that claims remain grounded in participant narratives rather than theoretical extrapolation.

Overall, these revisions primarily involve clarification and reframing, not additional data collection. Addressing them will significantly strengthen the manuscript’s methodological transparency and interpretive precision.

Reviewers' comments:

Reviewer's Responses to Questions

Comments to the Author

1. Is the manuscript technically sound, and do the data support the conclusions?

Reviewer #1: Partly

2. Has the statistical analysis been performed appropriately and rigorously? 

Reviewer #1: N/A

3. Have the authors made all data underlying the findings in their manuscript fully available?

Reviewer #1: Yes

4. Is the manuscript presented in an intelligible fashion and written in standard English?

Reviewer #1: Yes

5. Review Comments to the Author

Reviewer #1: This study addresses an underexplored area in higher education: the lived experience of teamwork among neurodivergent students. Despite its strengths, the study included participants who self-reported as neurodivergent, although 50% had a credible diagnosis. The study should have included all the participants with an established diagnosis.

Second, while the study integrates a conflict management framework, it does not include a neurotypical comparison group, which makes it difficult to determine which experiences are unique to neurodivergent students as compared to neurotypicals. This may limit causal or contrastive interpretation

6. PLOS authors have the option to publish the peer review history of their article (what does this mean?). If published, this will include your full peer review and any attached files.

Do you want your identity to be public for this peer review? For information about this choice, including consent withdrawal, please see our Privacy Policy.

Reviewer #1: No

---

## [Author Response · Author response to Decision Letter 1]

19 Feb 2026

We are grateful for the opportunity to revise and resubmit, and appreciate the feedback provided by the editor and reviewer.

In the email we received on January 5, 2026, additional requirements were outlined. Specifically, we were asked to update contact point information for data access. Since our data will not be made available under any circumstances, to protect the privacy of participants, we have removed contact information altogether.

Furthermore, comments implied that we will make our data available upon acceptance. Our data will not be made available. We updated the tick boxes under "Additional data availability information" in an effort to make this constraint more clear.

Finally, we were instructed to make changes to our financial disclosure in an updated cover letter, which we've attached.

Thank you for your consideration and time!

---

## [Editor Report · Decision Letter 1]

10 Mar 2026

Understanding Teamwork Experiences of Neurodivergent Students: A Phenomenological Exploration of Conflict and Collaboration in Engineering Teams

PONE-D-25-53622R1

Dear Dr. Sperry,

We’re pleased to inform you that your manuscript has been judged scientifically suitable for publication and will be formally accepted for publication once it meets all outstanding technical requirements.

Kind regards,

Ramandeep Kaur

Academic Editor

PLOS One
---

## [Editor Report · Acceptance letter]

PONE-D-25-53622R1

PLOS One

Dear Dr. Sperry,

I'm pleased to inform you that your manuscript has been deemed suitable for publication in PLOS One. Congratulations! Your manuscript is now being handed over to our production team.

Kind regards,

on behalf of

Dr. Ramandeep Kaur

Academic Editor

PLOS One